# RDNet: Rate–Distortion-Based Coding Unit Partition Network for Intra-Prediction

Chao Yao [1,†] , Chenming Xu [2,†] and Meiqin Liu [2,*]

1    School of Computer and Communication Engineering, University of Science and Technology Beijing, Beijing 100083, China; yaochao@ustb.edu.cn
2    Institute of Information Science, Beijing Jiaotong University, Beijing 100044, China; 21120329@bjtu.edu.cn
*    Correspondence: mqliu@bjtu.edu.cn
†    These authors contributed equally to this work.

**Abstract:** High efficiency video coding (HEVC) has been finalized as the most widely utilized video coding standard, jointly developed by ITU-T, VCEG, and MPEG. In HEVC, the quad-tree structure of the coding unit partition is one of the most substantial modules and provides significant coding gains following huge coding time. In this paper, a rate–distortion-based coding unit partition network (RDNet) is proposed to make partition decisions based on the statistical features. RDNet is composed of a prediction sub-network and a target sub-network, where the prediction sub-network is used to predict the CU partition modes of the intra-prediction and the target sub-network is designed to optimize the network parameters by evaluating the rate–distortion cost, respectively. To balance the prediction accuracy and the rate–distortion loss, a parameter-exchanging strategy is applied to control the parameters' sharing between two networks. Experimental results prove that our model can reduce the encoding time of HEVC by 55.83∼71.72% with an efficient BD-BR of 2.876∼3.347%, and the ablation study evaluates the ability of our strategy on balancing the trade-off between coding accuracy and inference speed.

**Keywords:** video coding; H.265/HEVC; coding unit partition; rate–distortion; intra-coding

## 1. Introduction

Video coding [1] is a key technique for multimedia applications that aims to reduce the bit-rate while preserving the critical visual information of video signals. In the past 30 years, a series of standards regarding video coding has been developed, such as MPEG2 Video/H.262 [2], H.264/AVC [3], H.265/HEVC [4], and H.266/VVC [5]. A variety of new coding techniques have been adopted, such as the quad-tree structure of coding unit (CU) partition and the affine motion compensation prediction. These techniques have dramatically improved the coding efficiency; however, the coding time of the codec has also been much higher. Specifically for HEVC and VVC, the recursive CU partition method [6] based on quad-tree takes more than 80% of the coding time; therefore, it is crucial to significantly reduce the coding time of HEVC, while maintaining the desirable coding efficiency.

Many attempts have been made to address the increasing coding time of the HEVC and VVC encoders. In general, these research works can be mainly classified into two categories: heuristic approaches and learning-based approaches. The heuristic strategies [7–12] mainly explore some heuristic features to skip the rate–distortion (RD) computation. The features are usually utilized to determine the CU partition in the early stage, for example, some textural homogeneity and spatial correlation are exploited to build statistical models for the CU partition. With these models, the rate–distortion optimization (RDO) computation are saved; however, the heuristic features cannot perfectly replace the role of RD, and the representation of heuristic features need to be also enhanced to assist the decision of CU partition. Alternatively, some approaches are further proposed to improve the efficiency of

the RD search by optimizing the traversal prediction mode. Unfortunately, there is still a cost associated with the RD search, so the reduced coding time is only marginal.

In these few years, the learning-based approaches have also been widely used in CU partition prediction. Driven by sufficient data, the planning of the CU partition can be automatically learned. Some machine learning methods, such as SVM [13], decision tree [14], and random forest [15], etc., are applied to analyze statistical features, then off-line pre-partition models are used to make decisions on the partition of CU Blocks. In contrast to heuristic approaches, learning-based approaches can address the drawback of heuristic feature extraction. Some of the latest works [16–21] have also introduced convolution neural networks (CNN) to learn CU partition, which can automatically extract deep features from large-scale images. It is noted that these works regard the CU partition as a binary classification problem, ignoring the impact of RD; therefore, the coding performance of these methods is generally limited.

In this paper, we propose a rate–distortion-based coding unit partition network (RD-Net) to improve the efficiency of CU partition of intra-coding in HEVC. The proposed approach designs a dual-network structure to jointly learn the statistical features and decide the partition modes of CU based on RD cost, which aims to optimize both coding time and RD performance in CU partition. To implement the optimization of the designed dual network, a parameter exchanging strategy is applied to balance the accuracy of CU partition towards RD performance and the corresponding RD loss. Experiment results show that the coding time of the proposed approach is greatly reduced compared with the standard partition approach in HEVC, with better compression visual effect on BD-BR and BD-PSNR.

To sum up, the main contributions of this paper are as follows:

- We design an RDNet that integrates a prediction network and a target network, to predict the possible CU splitting modes and the RD cost.
- We propose a parameters exchanging strategy to balance the accuracy of the CU partition and the RD cost. Meanwhile, a dynamic threshold is optimized to realize the rapid optimization of the network.
- We achieve a coding time reduction of 55.83~71.72% with an efficient BD-BR of 2.876~3.347%, compared to the HEVC test model (HM16.5).

## 2. Related Work

In order to develop fast video coding technology, many works have made tremendous efforts to optimize the CU partition.

### 2.1. Heuristic CU Partition

As shown in Figure 1, a classical coding tree unit (CTU) of one frame in the HEVC codec either contains a single CU or is recursively split into smaller square CUs via the quad-tree. The maximum size of a CTU is $64 \times 64$, and the minimal size of a CU can be $8 \times 8$. To determine whether a CU should be split up, RD cost is the most common metric. Due to the recursive partition along with the quad-tree, the CU partition operation consumes the largest fraction of the encoding time in the HEVC codec; therefore, the CU partition strategy determines the RD performance and time consumption.

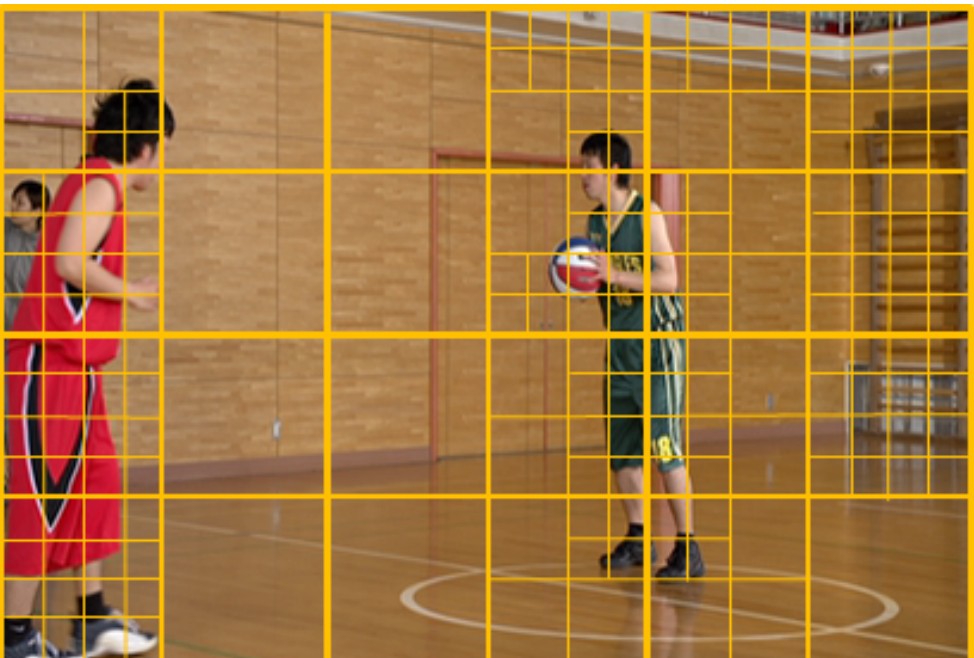

**Figure 1.** The results of CU partition. CU partition is a recursive process, which means all partitions are made before and RD is recursively traversed for merging.

To reduce the time consumption of CU partition, some approaches focus on the early decision of the CU partition, and make a lot of efforts to optimize the decision-making mode. Zhang et al. [7] utilized the entropy model to replace the RD optimization in the HEVC to skip the RD calculation in the encoding part. Zhang et al. [22] proposed to simplify the prediction model to improve the coarse RD search process, then implemented the acceleration of the CU partition. Lu et al. [23] proposed an intra-frame CU splitting method to predict the current CU partition by referring to the partition modes of the adjacent CUs. Jamali et al. [24] presented an RD cost prediction strategy based on a low-complexity statistics in the intra mode of HEVC for partition early termination. Zhang et al. [9] presented a hierarchical binary CU depth decision structure, which could balance the trade-off between the coding time and the RD performance. Similarly, Yang et al. [11] proposed a cascade decision optimized structure for the CU partition in the VVC. Wang et al. [25] optimized the original quad-tree structure and proposed an extended quad-tree to store the partition results. On the other hand, some heuristic features are utilized to assist the decision making, Kim et al. [8] and Fu et al. [12] proposed the online learning method to decrease the consuming time of CU partition, respectively. Qing et al. [10] proposed to predict the CU partitioning depth by calculating the corresponding salient features of the given image, in order to reduce the unnecessary partitioning modes. Kibeya et al. [26] proposed a matrix-based feature learning algorithm for judging the partition results in the early stage. Especially, Ferroukhi et al. [27] provided a method combining bandelets and the SPIHT based on 2nd-generation wavelets for CU partition in medical video coding. All the above works were able to reduce the coding time of HEVC up to a point, with the similar compression performance; however, the RD search is not skipped by using these methods, so that the time consumption of RD computation cannot be saved.

### 2.2. Learning-Based CU Partition

To tackle this problem, the learning-based approaches [8,28–30] introduce some statistical models trained from a large number of data to directly make decisions on the CU partition so that the recursive search of RD can be saved and the calculation time can be further reduced. Du et al. [28] proposed to apply random forest to skip or terminate CU partition in advance. Zhang et al. [29] utilized decision tree to predict the strategy of CU splitting offline. Then, an approach jointly using the online and offline mode was described

in [8]. Fu et al. [30] further presented the idea to extract the texture features from the input images, and support vector machines (SVM) were employed to split CU. To sum up, learning-based approaches improve the efficiency of the CU partition, but some limitations due to the representation of heuristic features still exist.

Recently, some approaches based on deep learning [16–21] have appeared to address the problem of CU partition. To enhance the representation of heuristic features, convolutional neural networks (CNN) are adopted to automatically extract features related to the CU partition. Liu et al. [16] first applied CNN to realize the optimization of HEVC in hardware and software. Jin et al. [17] utilized CNN to optimize the quad-tree of CU splitting. Wang et al. [19] designed a fast partition decision algorithm with a CNN-based quad-tree structure of CU. Xu et al. [18] designed an ETH-CNN to predict the CU partition, aiming to reduce the coding time of both the intra and the inter-mode in the HEVC. Zhang et al. [20] explored the texture information of the given images by using a CNN model to accelerate the CU partition. A similar work [21] continued to explore by applying the CNN approach on how to utilize the texture information to implement the fast decision of CU partition. Moreover, Feng et al. [6] utilized the depth map prediction for fast block partitioning based on CNN. Tissier et al. [31] trained CNN to predict a probability vector that speeds up coding block partitioning in VVC. Nevertheless, these works commonly regard CU partition as a binary classification problem, which leads to the binary output without information relating to RD cost; therefore, we propose a dual network based on RD, with the purpose of making some attempts introduce the RD cost in the training process.

## 3. Methodology

To introduce RD cost to balance the coding performance and speed, our RDNet is designed based on the dual network architecture, which is composed of a CU splitting prediction network and an RD target network. The original CTU and its corresponding mask, which is averagely split with $8 \times 8$, are utilized as inputs of RDNet. The prediction network consists of 3 convolutional layers and 1 full-connection (fc) layer, which is trained by a binary classification loss. In particular, the output of the prediction network is a hierarchical CU partition map (HCPM) as [18], which represents the CU partition mode. The architecture of the target network is similar to that of the prediction network, and the network is trained by an RD loss. To jointly train the dual network, parameters are shared between the target network and the prediction network. Consequently, we design a parameter exchanging strategy. Following with the fc layer of the target network, an RD memory is constructed to store and evaluate the calculated RD cost. In the parameters exchanging, parameters of the prediction network remain unchanged if and only if the RD cost is less than the previous value, otherwise the parameters are updated by those of the target network. The overall structure of RDNet is shown in Figure 2.

### 3.1. The RD Memory

The existing CNN-based approaches barely consider the RD cost during deciding on the CU partition, which makes it difficult to optimize the distortion. To resolve this issue, an RD memory is designed to store and evaluate the RD cost of the output HCPM from the prediction network in our approach. Correspondingly, a parameter-sharing strategy is proposed to make a balance between the target network and the prediction network.

As the input of the RD memory, HCPM is employed as the storage architecture to quantify the partition results of prediction network. Compared with the traditional CU partition in HEVC, which needs top-down computation for each layer, HCPM restores the total partition results of current CTU as a whole directly. It is convenient to measure the performance of prediction network and the ground truth. The architecture of ground truth (GT) HCPM and predicted HCPM is shown in the two parts of Figure 3. Specifically, different partition levels in HCPM have corresponding splitting depth $k$ (when $k$ is 1, it means that the CU is split to $32 \times 32$, and $k \in \{1, 2, 3\}$). The $y_1(U), y_2(U_i), y_3(U_{i,j}) \in \{0, 1\}$ and $\hat{y}_1(U), \hat{y}_2(U_i), \hat{y}_3(U_{i,j}) \in [0, 1]$ represent the partition confidence of CUs in k level for

GT HCPM and predicted one. In GT HCPM, $y_k(U) = 1$ represents the current CU will be split to next depth, and if $y_k(U) = 0$, the CU remains unchanged. Moreover, $\hat{y}_k(U)$ restores the partition confidence in predicted HCPM, which takes the binary classification threshold $thr_k$ as the line between non-partition and partition.

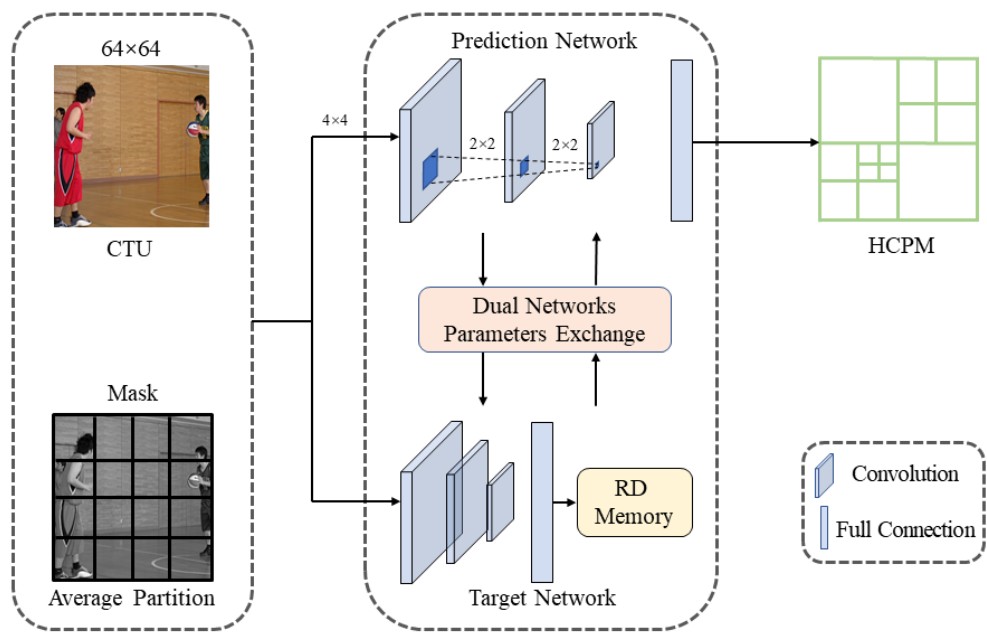

**Figure 2.** The overall pipeline of RDNet. The average partition mask is realized on the luminance matrix of original CTU. In addition, the convolution in prediction and target network contains one layer of $4 \times 4$ convolution and two layers of $2 \times 2$ convolution. The target network judges the best parameters by the RD memory, and optimizes the prediction network with the combination of the RD and classification loss.

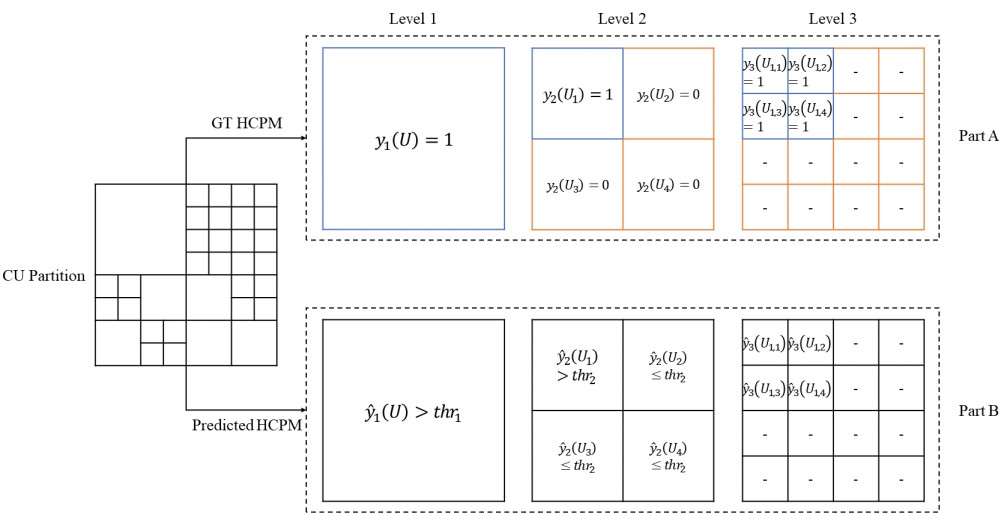

**Figure 3.** Structure of HCPM. The truth partition is shown in Part A. $y_1(U), y_2(U_i), y_3(U_{i,j})$ is the ground truth partition of each layer CUs. The subscripts $i, j \in 1, 2, 3, 4$ are the indices of the sub-CUs split from $U, U_i, U_{i,j}$. In level 1, If $y_1(U) = 1$ represents the CU will be split into four $32 \times 32$ CUs, conversely $y_1(U) = 1$ represents it will not be split. In Part B, the same CTU will obtain the corresponding HCPM prediction through network process. $\hat{y}_1(U), \hat{y}_2(U_i), \hat{y}_3(U_{i,j})$ represents the partition prediction value of each layer, and the partition prediction is obtained by the prediction network.

As shown in Figure 4, the target network is optimized by minimizing the difference between the predicted HCPM and the GT HCPM, where the ground truth HCPMs are obtained from the CU partition map in the HEVC encoder. RD memory of the target network is utilized to compute the RD distortion cost $Q_{RD}^k$ and the time cost $Q_T^k$ when the current CU is split to $64 \times 64$, $32 \times 32$ or $16 \times 16$. To balance the RD distortion cost and the time cost in the training duration, we use the total cost $Q^k$ to optimize the parameters of the target network, where $Q^k$ is the weighted sum of $Q_{RD}^k$ and $Q_T^k$:

$$Q^k = Q_T^k + \lambda Q_{RD}^k, \tag{1}$$

where $\lambda$ is a Lagrange constant, which locates in [0.5, 2].

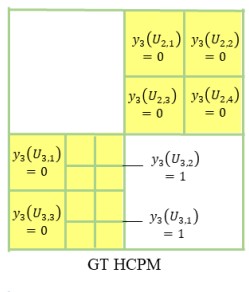

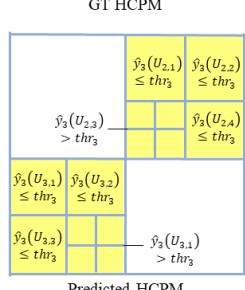

GT HCPM

Predicted HCPM

| $matrix^k$ | | GT | |
|---|---|---|---|
| | | Non-Partition | Partition |
| $k = 1$ Prediction | Non-Partition | 0 | 0 |
| | Partition | 0 | 1 |
| $k = 2$ Prediction | Non-Partition | 2 | 0 |
| | Partition | 0 | 2 |
| $k = 3$ Prediction | Non-Partition | 5 | 1 |
| | Non-Partition | 1 | 1 |

| | $Q_T^k$ | $Q_{RD}^k$ | $Q^k$ |
|---|---|---|---|
| $k = 1$ | 1 | 0 | 1 |
| $k = 2$ | 2 | 0 | 2 |
| $k = 3$ | 1 | 2 | 5 |

**Figure 4.** An example of the RD memory computation. The RD memory aims to record the difference between ground truth HCPM and network results. The corresponding partition results of different levels of CU blocks are recorded in $matrix^k$. $Q^k$ can be calculated from $matrix^k$ for three partition levels.

As it is difficult to measure the real RD cost and the coding time during training stage, we adopt a parametric strategy to simulate RD cost and coding time in the RD memory. Given the GT HCPM and the predicted HCPM, a $2 \times 2$ matrix $matrix^k[m][n]$ is applied to store the partition decisions, where $m$ represents the GT partition decision from HEVC and $n$ denotes the result of the predicted HCPM, $m, n \in [0, 1]$. Moreover, the value 1 store in the matrix indicates to perform the partition, and 0 represents non-partition. Then, $Q_{RD}^k$ is employed to evaluate the different partition decision between the prediction HCPM and the GT one, as:

$$Q_{RD}^k = matrix^k[0][1] + matrix^k[1][0] \tag{2}$$

In addition, the time cost $Q_T^k$ is used to count the number of CUs, which has to do with the partition, as:

$$Q_T^k = matrix^k[0][1] + matrix^k[1][1] \tag{3}$$

Some details about an example of applying the defined matrix for calculating the RD cost are shown in Figure 4. First, a predicted HCPM and the corresponding GT HCPM are given as input, $matrix^k$ records the different partition situations. The yellow part marks the partition level of $k = 3$ in both the predicted HCPM and the GT one, this means the current CU should be split to $8 \times 8$ CUs, where the number of CUs without partition of both HCPMs is 5, which are $U_{2,1}, U_{2,2}, U_{2,4}, U_{3,1}, U_{3,3}$, respectively, and the number of CU with partition for both HCPMs is 1. Further, there are 2 CUs with different results of both HCPMs as $U_{2,3}, U_{3,2}$. Then, $Q_T^3$ and $Q_{RD}^3$ can be computed as Equations (2) and (3). When

$\lambda = 2$ in Equation (1), $Q^3 = 5$ in the RD memory. Similarly, $Q^1 = 1$ and $Q^2 = 2$ are obtained by the same strategy.

### 3.2. Parameters Exchanging Strategy

RDNet integrates two networks to predict the partition maps and compute the RD difference. For the further optimization of these two networks, we design a parameter-exchanging strategy. In order to share the parameters between two networks in the training process, we set that if and only if $Q^k$ is less than the previous stored value in the memory, the parameters of the prediction network can remain unchanged, otherwise they are updated by those of the target network. Moreover, to maximize the accuracy of the binary classification and minimize the RD cost, a total loss function is applied to jointly optimize the whole dual network, which includes a classification loss and a defined RD loss, as

$$\mathcal{L}^k = \mathcal{L}^k_c + \mathcal{L}^k_{RD}, \tag{4}$$

where $\mathcal{L}^k$ is the total loss and $k$ represents the partition level, $\mathcal{L}^k_c$ is the classification loss to sum the cross-entropy over all valid elements of GT HCPM and predicted HCPM, the same as the binary classification in [18], as

$$\mathcal{L}^k_c = \begin{cases} H(y_1(U), \hat{y}_1(U)) & k = 1 \\ \sum\limits_{\substack{i \in \{1,2,3,4\} \\ H(y_2(U_i)) \neq \varnothing}} H(y_2(U_i), \hat{y}_2(U_i)) & k = 2 \\ \sum\limits_{\substack{i,j \in \{1,2,3,4\} \\ H(y_3(U_{i,j})) \neq \varnothing}} H(y_3(U_{i,j}), \hat{y}_3(U_{i,j})) & k = 3 \end{cases} \tag{5}$$

where $H(\cdot, \cdot)$ denotes the cross-entropy between the elements of ground truth and predicted HCPM. $\mathcal{L}^k_{RD}$ is the RD loss, which is represented as the $\mathcal{L}_1$ loss between the RD of current predicted HCPM and best RD in the memory, as

$$\mathcal{L}^k_{RD} = |Q^k - Q^k_{best}| \tag{6}$$

where $Q^k$ is the RD of current predicted HCPM, $Q^k_{best}$ is the stored best RD in the RD memory.

### 3.3. Fast Partition Neural Network

In general, neural networks are utilized to learn the context features and making partition decision, such as [18]; however, to skip the CU partition of the current frame in the HEVC Test Model (HM) [32], it is still necessary to read the existing partition model that has been obtained by neural network in the coding process; therefore, the HCPM introduced in Section 3.1 is employed as an output of the network to skip the recursion computation in HEVC.

In order to further speed up the process of the neural network, a pre-processing layer is adopted to perform down-sampling with average pooling on the CTU original luminance matrix, as shown in Figure 5. Due to the fact that HCPM records the CU partition results of different levels, the corresponding layers in the neural network should be also matched with three branches $B_1, B_2, B_3$. In our prediction network, each branch uses 3 convolutional layers to extract feature maps, and merges the obtained features into a vector before entering in fc. Towards justifying the spatial detail compression quality of current CU, the vector is further divided into three channels again to connect with the quantization parameter (QP) [33], where lower value means the finer details preserving and higher value denotes the lost about some details. Then, $\mathcal{L}^k$ is utilized in corresponding branches to optimize the different branches of the network and deal with the different levels CU partition. Following with the network, it can obtain a value that belongs to [0,1] for the partition decision. The current CU will be split if the value is greater than 0.5. Since

the RD memory and parameter-exchanging strategy change the optimization direction of network, a dynamic threshold strategy is also designed.

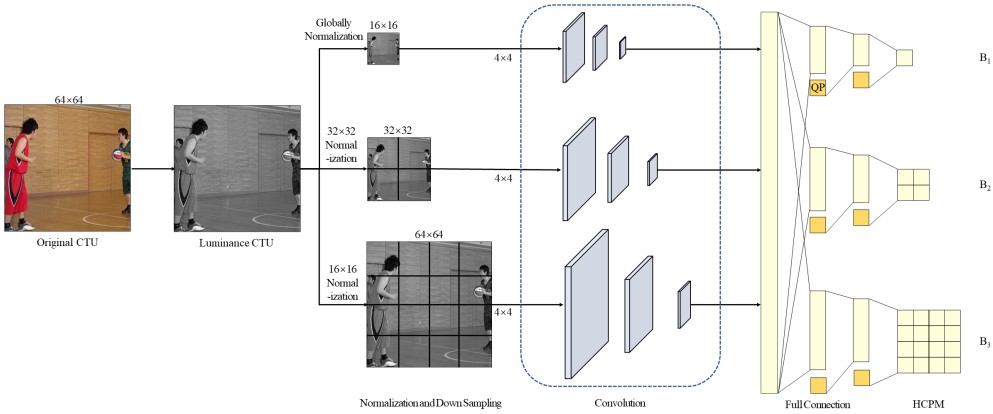

**Figure 5.** The structure of CNN. The network mainly includes three parts, pre-process, convolution, and decision making. The CTU are demeaned and down-sampled to reduce the computation time in pre-processing, the convolution includes 3 convolutional layers and an fc layer to extract image features, and the decision-making part judges the CU partition to HCPM by features.

To make the threshold approach the appropriate value, it needs to reduce the possible mistakes in the partition results. Similarly with $Q_{RD}^k$, $matrix^k[0][1]$ and $matrix^k[1][0]$, which represents the wrong partition, are utilized in the threshold seeking. The $matrix^k[1][0]$ indicates the excessive partition of the predicted HCPM, which means it need to raise the current threshold for stopping this partition. On the contrary, $matrix^k[0][1]$ indicates the threshold of the network is higher than the suitable value. So the threshold requires adjusting after meeting the corresponding mistakes starting from 0.5. The final threshold can be set as:

$$thr_k = 0.5 + \gamma(matrix^k[0][1] - matrix^k[1][0]) \tag{7}$$

where $thr_k$ represents the final threshold for different levels, $\gamma$ are the scale factors of $matrix_k[0][1]$ and $matrix_k[1][0]$.

## 4. Experiment Results

In this section, we train the RDNet for the CU partition of the intra-mode, and validate by applying in the HM16.5, which is utilized in previous works [18,34] as the evaluation benchmarks. The ablation experiments are also designed to evaluate each designed component in our framework.

### 4.1. Configuration and Settings

The training data used in the experiment is CPH-intra [35] data set, which comes from 2000 lossless images and is compressed by four different QPs. Each sample is composed of the luminance matrix of the CU and a ground truth binary label representing whether to partition or not. The test data are five different types of JCT-VC standard test sequences [32]. The data properties used in ablation study are shown in Table 1. In the final experimental evaluation, we implement more test sequences with different resolutions from JCT-VC for fair comparison, all settings follow the default in CTC [32].

**Table 1.** Partly sequences of the JCT-VC test set.

| Class | Resolution Ratio | Test Sequence | Total Frames | FPs | Bit Depth |
|-------|------------------|---------------|--------------|-----|-----------|
| A | 2560 × 1600 | *Traffic* | 150 | 30 | 8 |
| B | 1920 × 1080 | *Basketball Drive* | 500 | 50 | 8 |
| C | 832 × 480 | *Basketball Drill* | 500 | 50 | 8 |
| D | 416 × 240 | *Basketball Pass* | 500 | 50 | 8 |
| E | 1280 × 720 | *Johnny* | 600 | 60 | 8 |

For specific optimization details, we use momentum optimizer and the learning rate is set to 0.01 with TensorFlow-1.14. The parameters $\gamma$ is 0.001 in Equation (7). The neural network trains on one 11GB GTX 1080Ti. The intra-mode encoder used in the experiments is HM 16.5, testing on Intel(R) Core(TM) i7-8750H CPU with 8 GB RAM. Since all the decisions are made in the intra-coding, so in HM 16.5, the AI configuration was applied with the default configuration file encoder_intra_main.cfg for the performance evaluation.

In our experiments, the Bjøntegaard delta bit-rate (BD-BR), Bjøntegaard delta PSNR (BD-PSNR), and $\Delta T$ in VCEG-M33 [36] are utilized to objectively evaluate the current experiment results and to represent the coding performance difference between all optimized approaches and standard coding HM16.5. BD-BR represents the bit rate increment of the optimized algorithm compared with the original algorithm under the condition of the same objective video quality, as:

$$BD-BR = \frac{1}{D_H - D_L} \int_{D_L}^{D_H} (r_2 - r_1) dD \tag{8}$$

where $D_H$, $D_L$ are high and low ends of the output RD curves range, respectively, and $r_1$, $r_2$ are two corresponding bit rates. BD-PSNR represents the objective video quality improvement of the optimized algorithm compared with the original algorithm at the same bit rate, as:

$$BD-PSNR = \frac{1}{r_H - r_L} \int_{r_L}^{r_H} (D_2(r) - D_1(r)) dr \tag{9}$$

where $r_H = log(R_H)$, $r_L = log(R_L)$ are high and low ends of the output bit rate range, respectively, and $D_1(r)$, $D_2(r)$ are two corresponding RD curves. $\Delta T$ represents the time saved by the optimized algorithm in the case of coding the same number of frames. The calculation method of $\Delta T$ is shown in Equation (10):

$$\Delta T = \frac{(T' - T)}{T} \times 100\% \tag{10}$$

where $T$ is the coding time consumed by standard coding, $T'$ represents the coding time consumed by the optimized algorithm.

### 4.2. Ablation Study

In order to evaluate the performance of RDNet and dynamic threshold, we complete four groups of experiments in ablation study whose detail are shown in Table 2. The four groups of ablation experiments use different experiment settings. To be specific, the ETH-CNN with the partition threshold is adopted as the backbone of the ablation study. RDNet-$E_1$ and RDNet-$E_2$ are used to represent the ablation of different modules, and RDNet-$E_3$ means the final proposed strategy. The dynamic threshold and the RD-based dual network are used for ablation analysis. The threshold of experiments without dynamic threshold is maintained as [0.5 0.5 0.5]. The threshold of experiments with dynamic threshold maintains is adjusted by the threshold training strategy in Equation (7) and the results are shown in Table 2. The model results needs to be tested in four different QPs (QP = 22, 27, 32, 37) with the first 20 frames of each sequence. The test results are compared by calculating BD-BR and BD-PSNR in VCEG-AE07.xls [32]. The calculation results are shown in Table 3.

**Table 2.** The detail of the four groups ablation experiments.

| Algorithm | Dynamic Threshold | RD-Based Dual Networks | Partition Threshold |
|---|---|---|---|
| ETH-CNN [18] | - | - | [0.50 0.50 0.50] |
| RDNet-$E_1$ | ✓ | - | [0.49 0.55 0.63] |
| RDNet-$E_2$ | - | ✓ | [0.50 0.50 0.50] |
| RDNet-$E_3$ | ✓ | ✓ | [0.48 0.55 0.63] |

Due to the fact that the fast algorithm is used to skip the recursive RD calculation process in HEVC, the above approaches have a certain impact on the compression performance in the case of large-scale reduction of computation time. The BD-BR values in the table are all positive, which means that the compression performance is not as good as HM16.5; however, the approach with the lower BD-BR has the better coding performance. The BD-PSNR values are all negative in the table, which means that the video coding quality is not as good as HM16.5; in contrast, the approach that has the higher BD-PSNR has the better coding quality. The specific relationship between BD-BR and BD-PSNR from Table 3 is shown in Figure 6.

**Table 3.** The ablation study results of JCT-VC test set (AI).

| Class | Test Sequence | Algorithm | BD-PSNR (dB) | BD-BR (%) | $\Delta T$ (%) | | | |
|---|---|---|---|---|---|---|---|---|
| | | | | | QP = 22 | QP = 27 | QP = 32 | QP = 37 |
| A | *Traffic* | ETH-CNN [18] | −0.149 | 2.771 | −56.62 | −65.17 | −67.79 | −63.16 |
| | | RDNet-$E_1$ | −0.133 | 2.480 | **−60.44** | **−66.98** | **−70.71** | **−69.01** |
| | | RDNet-$E_2$ | −0.148 | 2.757 | −59.57 | −66.41 | −70.27 | −67.80 |
| | | RDNet-$E_3$ | **−0.131** | **2.429** | −58.13 | −63.82 | −68.29 | −63.96 |
| B | *Basketball Drive* | ETH-CNN [18] | −0.119 | 4.981 | −57.90 | −72.74 | −79.33 | -79.18 |
| | | RDNet-$E_1$ | **−0.094** | **3.904** | **−67.05** | **−78.18** | −81.53 | −81.63 |
| | | RDNet-$E_2$ | −0.12 | 4.967 | −61.00 | −76.26 | **−81.62** | **−82.37** |
| | | RDNet-$E_3$ | **−0.094** | 3.941 | −61.26 | −75.76 | −78.70 | -81.43 |
| C | *Basketball Drill* | ETH-CNN [18] | −0.141 | 2.934 | −23.34 | −36.87 | −52.66 | −62.19 |
| | | RDNet-$E_1$ | −0.134 | 2.796 | **−35.75** | **−50.64** | −57.38 | −64.14 |
| | | RDNet-$E_2$ | −0.142 | 2.969 | −30.80 | −47.48 | −57.37 | −64.48 |
| | | RDNet-$E_3$ | **−0.130** | **2.738** | −20.33 | −47.09 | **−57.90** | **−66.16** |
| D | *Basketball Pass* | ETH-CNN [18] | −0.138 | 2.412 | −40.54 | −40.30 | −54.42 | −58.00 |
| | | RDNet-$E_1$ | −0.116 | 2.029 | -40.90 | −44.17 | −56.06 | −60.99 |
| | | RDNet-$E_2$ | −0.135 | 2.359 | −37.33 | −43.29 | −50.66 | −47.70 |
| | | RDNet-$E_3$ | **−0.107** | **1.853** | **−49.72** | **−53.74** | **−59.48** | **−65.29** |
| E | *Johnny* | ETH-CNN [18] | −0.146 | 3.636 | −73.62 | −73.54 | −74.45 | −78.55 |
| | | RDNet-$E_1$ | **−0.136** | **3.355** | −74.98 | −76.49 | −77.97 | −80.58 |
| | | RDNet-$E_2$ | −0.141 | 3.501 | −74.37 | −76.03 | **−80.71** | **−82.81** |
| | | RDNet-$E_3$ | -0.138 | 3.421 | **−75.25** | **−77.21** | −76.52 | −81.23 |
| | Std. dev. | ETH-CNN [18] | **0.011** | 1.016 | 19.13 | 17.81 | 11.88 | 9.91 |
| | | RDNet-$E_1$ | 0.018 | **0.735** | **16.88** | 15.29 | 11.64 | 9.43 |
| | | RDNet-$E_2$ | **0.011** | 1.013 | 18.04 | 15.66 | 13.84 | 14.54 |
| | | RDNet-$E_3$ | 0.018 | 0.821 | 20.42 | **13.26** | **9.51** | **8.90** |
| | Best | ETH-CNN [18] | −0.119 | 2.412 | −73.62 | −73.54 | -79.33 | −79.18 |
| | | RDNet-$E_1$ | **−0.094** | 2.029 | −74.98 | **−78.18** | −81.53 | −81.63 |
| | | RDNet-$E_2$ | −0.120 | 2.359 | −74.37 | −76.26 | **−81.62** | **−82.81** |
| | | RDNet-$E_3$ | **−0.094** | **1.853** | **−75.25** | −77.21 | −78.70 | −81.43 |
| | Average | ETH-CNN [18] | −0.138 | 3.347 | −50.40 | −57.73 | −65.73 | −68.22 |
| | | RDNet-$E_1$ | −0.123 | 2.913 | **−55.83** | −63.29 | **−68.73** | -71.27 |
| | | RDNet-$E_2$ | −0.137 | 3.311 | −52.61 | −61.90 | −68.13 | −69.03 |
| | | RDNet-$E_3$ | **−0.120** | **2.876** | −52.94 | **−63.52** | −68.18 | **−71.62** |

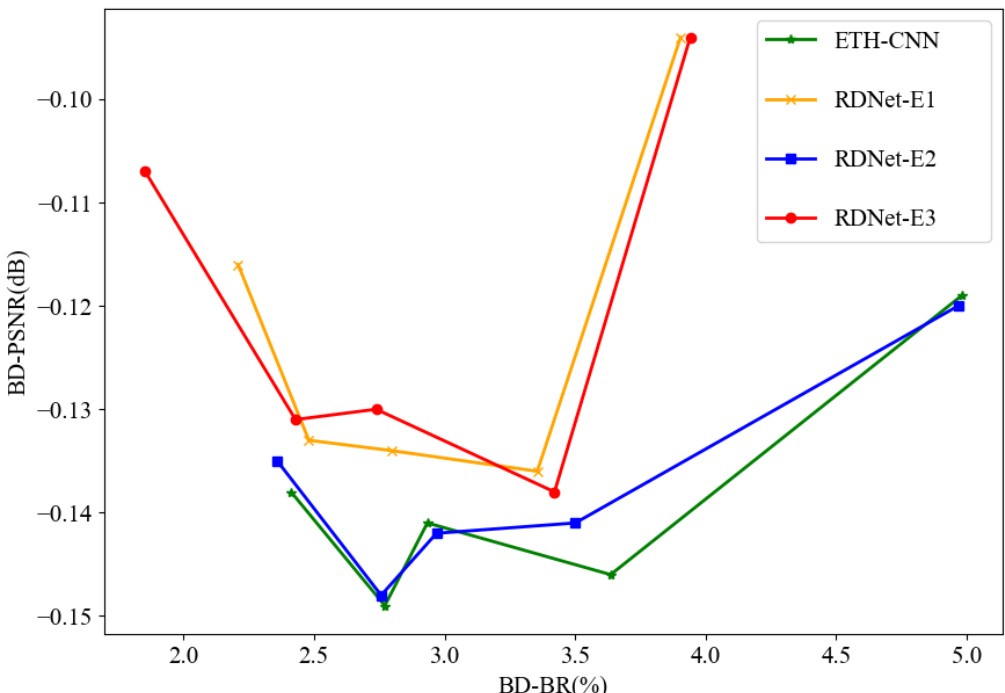

**Figure 6.** The ablation study results of JCT-VC test set (AI). The closer the data points are to the upper left corner of the graph, the higher the coding quality at a lower coding bit-rate, which means better coding performance. RDNet-$E_3$ in the figure is closest to the upper left corner, indicating the best coding performance. From the other three curves, it can be concluded that the proposed dual network and dynamic threshold can make the optimization to the traditional approaches.

From the calculation results in the above table, it can be concluded that the threshold judged by the prediction network is affected by the target network due to RDNet. In RDNet-$E_1$, the thresholds develop irreversibly, which results in poor coding performance. RDNet-$E_2$ shows that the dynamic thresholds can be adapted to the process of supervising optimization between dual networks. Without the dynamic thresholds, the CU partition of dual networks is not as good as that of a single network. Although RDNet-$E_3$ does not perform best in each resolution of the test sequence, it has excellent performance on average, suggesting that the approach adapts to the CU partition for various resolution test sequences. At the same time, it saves 64.06% time compared with HM16.5, improves the video coding compression efficiency in the intra-frame mode level, and proves the validity of the dual networks.

### 4.3. Performance Evaluation

In order to evaluate the proposed RDNet whose optimization performance for video coding with other classical efficient approaches, we compare our work to a fast algorithm FSD-SVM [34] based on heuristic features and a single network without RD approach PPMAC [16]. The experiment results are shown in Table 4.

From the comparison results in Table 4, it can be concluded that the proposed RDNet is outstanding in compression effect and time with better stability. To test sequence, the RDNet refers to the HEVC standard coding for realizing the supervision on the single network with RD. Compared with the depth modules without RD supervision, the RDNet has the better compression performance in intra-frame video coding.

**Table 4.** Performance comparison with state-of-the-art approaches used the same strategy.

| Class | Test Sequence | FSD-SVM [34] | | | PPMAC [16] | | | RDNet-$E_3$ | | |
|---|---|---|---|---|---|---|---|---|---|---|
| | | BD-PSNR (dB) | BD-BR (%) | $\Delta T$ (%) | BD-PSNR (dB) | BD-BR (%) | $\Delta T$ (%) | BD-PSNR (dB) | BD-BR (%) | $\Delta T$ (%) |
| A | PeopleOnStreet | −0.942 | 9.627 | −43.84 | −0.209 | 3.969 | −55.60 | −0.127 | **2.197** | −57.53 |
| | Traffic | −0.304 | 6.411 | −28.87 | −0.240 | 4.945 | −60.84 | −0.131 | **2.429** | −63.55 |
| B | BasketballDrive | −0.244 | 8.923 | −43.40 | −0.141 | 6.018 | −69.51 | −0.094 | **3.941** | −74.29 |
| | BQTerrace | −0.295 | 6.627 | −56.62 | −0.267 | 4.815 | −57.89 | −0.078 | **1.191** | −47.96 |
| | Cactus | −0.248 | 7.533 | −43.51 | −0.208 | 6.021 | −63.23 | −0.075 | **1.945** | −52.72 |
| | Kimono | −0.170 | 5.212 | −47.80 | −0.082 | 2.382 | −72.72 | −0.051 | **1.403** | −83.53 |
| | ParkScene | −0.149 | 3.630 | −52.85 | −0.135 | 3.417 | −66.03 | −0.076 | **1.756** | −59.25 |
| C | BasketballDrill | −0.439 | 9.818 | −53.93 | −0.538 | 12.205 | −63.58 | −0.130 | **2.738** | −47.87 |
| | BQMall | −0.486 | 9.646 | −42.06 | −0.468 | 8.077 | −52.14 | −0.084 | **1.333** | −33.08 |
| | PartyScene | −0.468 | 7.383 | −43.01 | −0.672 | 9.448 | −58.75 | −0.028 | **0.363** | −33.66 |
| | RaceHorses | −0.379 | 7.220 | −44.59 | −0.264 | 4.422 | −58.20 | −0.107 | **1.656** | −36.28 |
| D | BasketballPass | −0.546 | 10.054 | −39.72 | −0.457 | 8.401 | −63.53 | −0.107 | **1.853** | −57.06 |
| | BlowingBubbles | −0.373 | 6.178 | −37.04 | −0.463 | 8.328 | −60.78 | −0.052 | **0.845** | −37.87 |
| | BQSquare | −0.876 | 12.342 | −57.43 | −0.211 | 2.563 | −46.72 | −0.022 | **0.263** | −38.67 |
| | RaceHorses | −0.487 | 8.839 | −40.23 | −0.317 | 4.593 | −57.30 | −0.068 | **0.977** | −42.99 |
| E | FourPeople | −0.480 | 9.077 | −36.22 | −0.439 | 8.002 | −61.54 | −0.173 | **2.905** | −64.20 |
| | Johnny | −0.474 | 12.182 | −63.55 | −0.307 | 7.956 | −66.55 | −0.138 | **3.421** | −77.55 |
| | KristenAndSara | −0.627 | 13.351 | −57.51 | −0.265 | 5.478 | −64.72 | −0.139 | **2.662** | −74.00 |
| | Std. dev. | 0.175 | 2.553 | 9.02 | 0.158 | 2.603 | **6.17** | **0.041** | **1.008** | 15.97 |
| | Best | −0.149 | 3.630 | −63.55 | −0.082 | 2.382 | −72.72 | **−0.022** | **0.263** | **−83.53** |
| | Average | −0.419 | 8.559 | −46.23 | −0.316 | 6.189 | **−61.09** | **−0.093** | **1.883** | −54.56 |

## 5. Conclusions

In this paper, we proposed an RDNet to optimize the decision of CU partition based on RD cost. The approach introduces a sub-network to select the best parameters for the RD memory and predict the partition mode with dynamic threshold. Particularly, the RD memory in the target network is designed to evaluate the results of the prediction network with GT HCPM to keep the best parameters. To avoid deviation from the correct optimization direction, the dynamic threshold strategy is presented to adapt to the change regarding RD cost. Experimental results show the proposed method improves the quality and efficiency of the HEVC codec with a significant reduction in coding time.

**Author Contributions:** Conceptualization, C.Y.; methodology, C.Y. and C.X.; software, C.X.; validation, C.X., C.Y. and M.L.; formal analysis, C.X.; investigation, C.X.; resources, C.Y. and M.L.; data curation, C.X.; writing—original draft preparation, C.Y.; writing—review and editing, C.Y.; visualization, C.X.; supervision, C.X., C.Y. and M.L.; project administration, C.X., C.Y. and M.L.; funding acquisition, C.Y. and M.L. All authors have read and agreed to the published version of the manuscript.

**Funding:** This research was funded by the Fundamental Research Funds for the Central Universities (FRF-TP-19-015A1, FRF-IDRY-20-038) and the National Natural Science Foundation of China (61972028, 61902022).

**Conflicts of Interest:** The authors declare no conflict of interest.

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
