# Peer review of "RDNet: Rate–Distortion-Based Coding Unit Partition Network for Intra-Prediction"

_electronics, doi:10.3390/electronics11060916_

Round 1

Reviewer 1 Report

This paper proposes a learning-based CU partitioning scheme for H.256/HEVC.
With respect to other comparable approaches, the main novelty here seems to be sharing the parameters by the two networks doing the partitioning and predicting the encoding cost.
At the present time, it is not possibile to me to recommend the paper for publication due to the below reasons.

First, it is not clear wether the authors evaluated the proposed method over the > 30 standard jvet test sequences that are commonly used in similar experiments or just the 5 sequences in Table 1. The authors must clarify on this point and, if that is not the case, the authors must perform the missing experimenst with all the test sequences available at ftp://ftp.hhi.fraunhofer.de/ctc/sdr/.

Second, it is not clear what is the reference for measuring the savings in encoding time. Reading at Sec 4.2, it seems that in table 3 and following the reference is the HM16.5 reference encoder. Now, while this is the reference HEVC encoder, it is by no means meant to be optimized nor used as a benchmark for encoding time. The authors shall also report versus a more production-oriented encoder such as the opensource libx265 https://x265.readthedocs.io/

Third, about the complexity of thr proposed method, what is thr network complexity, eg in terms of number of memory footprint and processing capabilities (eg, number of MADDs/FMAs) required ? I see the authors benchmark ona  GTX 1080Ti, a server-grade GPU that drains over 100W of power when runnig at full spin. While this is perfectly fit for training and for producing RD performance numbers, that is a very optimistic reference for power consumption and running times. The authors are strongly encouraged to test also on a more embedded-oriented  architecture, e.g. an NVIDIA jetson board. Of course, complexity reducing strategies such as FP16 precision or network pruning may be taken into consideration.
Finally, the authors claim a "negligible BD-BR [loss] of 2.876% - 3.347%": while my understaniding is that these numbers relate to an all-intra scheme, these numbers cannot be labeled as negligible. Video traffic accounts nowadays > 60% of teh internet traffic according to CISCO reports available online, and 3% of that would stimm be > 1% ofthe overall internet traffic.

Reviewer 2 Report

As a general comment and request, please, explain if your CU partition proposal depends on the quantization parameter QP. Another important aspect that needs futher clarification is the computation environment used for the time comparison to previous proposals. Finally, the description of the proposal can be significatively improved.

Reviewer 3 Report

This manuscript deals with an interesting topic but has some shortcomings, notably a comparison with other recent methods. 
The case of highly noisy images has not been considered.
The manuscript would gain in readability by avoiding too much use of acronyms. In any case, all acronyms should be defined as soon as they are used, but even in the abstract BD-BR is not defined.
The authors should also analyse and cite the important recent paper on video coding.

Ferroukhi, M. et al. A. Medical Video Coding Based on 2nd-Generation Wavelets: Performance Evaluation. Electronics 20198, 88. 

Other shortcomings include a clear description of the architecture adopted and the data used for training and validation.

Round 2

Reviewer 1 Report

The authors have addressed the concerns I raised over the previosu round of reviews, so I can now recommend this paper for publication.

Reviewer 3 Report

The authors have generally taken our recommendations into account